Spatial and temporal distribution patterns of tick-borne diseases (Tick-borne Encephalitis and Lyme Borreliosis) in Germany

Cunze Sarah 1 cunze@bio.uni-frankfurt.de
Glock Gustav 1
Klimpel Sven 1 2
1 Institute of Ecology, Evolution and Diversity, Johann Wolfgang Goethe Universität Frankfurt am Main , Frankfurt am Main, Hesse , Germany
2 Biodiversity and Climate Research Centre, Senckenberg Nature Research Society , Frankfurt am Main, Hesse , Germany
Braga Erika
Electronic publication date: 2021 Dec 13
Publication date: 2021
Volume: 9
Electronic Location ID: e12422
Received 2021 Feb 25; Accepted 2021 Oct 11
Copyright: © 2021 Cunze et al.
Copyright year: 2021
Copyright holder: Cunze et al.
License: This is an open access article distributed under the terms of the Creative Commons Attribution License, which permits unrestricted use, distribution, reproduction and adaptation in any medium and for any purpose provided that it is properly attributed. For attribution, the original author(s), title, publication source (PeerJ) and either DOI or URL of the article must be cited.
License URL: https://creativecommons.org/licenses/by/4.0/

Keywords: Ixodes ricinus, Tick-borne diseases, Vector-host-interaction, Weather conditions, Land cover, Robert Koch Institute, Castor bean tick, Climate change

Funding: The authors received no funding for this work.

==============================
Background

In the face of ongoing climate warming, vector-borne diseases are expected to increase in Europe, including tick-borne diseases (TBD). The most abundant tick-borne diseases in Germany are Tick-Borne Encephalitis (TBE) and Lyme Borreliosis (LB), with Ixodes ricinus as the main vector.

Methods

In this study, we display and compare the spatial and temporal patterns of reported cases of human TBE and LB in relation to some associated factors. The comparison may help with the interpretation of observed spatial and temporal patterns.

Results

The spatial patterns of reported TBE cases show a clear and consistent pattern over the years, with many cases in the south and only few and isolated cases in the north of Germany. The identification of spatial patterns of LB disease cases is more difficult due to the different reporting practices in the individual federal states. Temporal patterns strongly fluctuate between years, and are relatively synchronized between both diseases, suggesting common driving factors. Based on our results we found no evidence that weather conditions affect the prevalence of both diseases. Both diseases show a gender bias with LB bing more commonly diagnosed in females, contrary to TBE being more commonly diagnosed in males.

Conclusion

For a further investigation of of the underlying driving factors and their interrelations, longer time series as well as standardised reporting and surveillance system would be required.

Introduction

Tick-borne diseases in Europe

In recent years, tick populations in Europe have been observed to expand their range further north to higher latitudes and altitudes (Gilbert, 2010a; Süss, 2011). This range shift has been accompanied by an increase in the number of human tick-borne disease (TBD) cases (Gray et al., 2009; Heyman et al., 2010). However, this increase could also be due, at least in part, to better diagnostics and higher awareness.

Ticks are competent vectors for a number of pathogens, including viruses, bacteria or parasites, which can cause serious infections in humans and animals. Ixodes ricinus is the most abundant and widespread tick species in Europe (Gray et al., 2009) and the primary vector for Lyme borreliosis (LB) and tick-borne encephalitis (TBE), the most common diseases associated to ticks in Europe (Semenza & Suk, 2018).

Lyme borreliosis, also known as Lyme disease, is caused by bacteria of the Borrelia burgdorferi sensu lato complex, which comprises about 20 members; six of them are confirmed to be human pathogens (Lin et al., 2020). With approximately 65,000 cases per year, LB is the most frequent tick-borne disease in the European Union (Robert Koch-Institut, 2020).

TBE is a viral infection caused by one of three tick-borne encephalitis virus subtypes belonging to the Flaviviridae family; over the last 30 years, a large increase of nearly 400% in the number of recorded cases has been observed worldwide (Fong, 2020). Although this increase can be partly explained by enhanced surveillance and diagnostic TBE should be considered a growing public health challenge in Europe and other parts of the world (Medlock et al., 2013).

Both pathogens are transmitted to humans through the bite of infected ticks.

Ecology and life cycle of Ixodes ricinus

Ixodes ricinus, the primary vector of TBE and LB in Europe, is a three-host tick: After hatching from the egg, there are three distinct stages (larva, nymph, adult) in the life cycle. Each passage, from larva to nymph as well as from nymph to adult stage, requires a blood meal before the subsequent moult. Ticks only feed once per life stage (larva, nymph and adult female). After the blood-sucking process, that takes about 2–4 days for larvae and 3–5 days for nymphs, the tick detaches and develops in the vegetation (moulting) (Gray, 1998). After mating and another blood meal (duration of about 6–10 days) the female adults tick lays a batch of between 1,000 and 5,000 eggs and then dies (Süss, 2003). The average timespan of each stage is approximately 8–12 months but can be extended to 2 up to 6 years (Gray, 1998; Stanek et al., 2012). Under favourable environmental conditions ticks are able to survive for several months without feeding (Heyman et al., 2010).

Like most ectothermic terrestrial organisms, the development and activity of ticks depend on weather and climatic conditions. Ixodes ricinus ticks may enter a prolonged behavioral or developmental diapause when hosts are not available or environmental conditions are unsuitable for the ticks (Gray, 1998; Heyman et al., 2010). The activity phase begins when temperatures rise to a certain level in spring (to 5 to 7 °C), i.e., March/April according (Süss, 2003) and ends when it gets too cold in late autumn or winter. Other factors, such as summer drought, can regionally lead to reduced tick activity. Mild winters or in thermally favoured regions can lead to activity during the winter (Dautel et al., 2008).

The host spectrum of I. ricinus comprises more than 300 vertebrate species (wild and domestic mammals, reptiles and birds). Preferred host species differ between the life stages (Stanek et al., 2012): The larvae feed on animals of all sizes (Süss, 2011), particularly small mammals, birds and reptiles, nymphs feed on medium-sized mammals such as foxes but also on birds and humans (Süss, 2011). The adult stage feeds more exclusively on larger mammals such as deer, sheep, dogs and humans (Medlock et al., 2013).

Ixodes ricinus shows an exophilic (non-nidicolous) behaviour in all stages, i.e., individuals do not live in or near the shelters used by their vertebrate hosts. Preferred habitats are all woodlands, especially deciduous forests and the transition zones between forests and grasslands (Estrada-Peña, Mihalca & Petney, 2018). Ixodes ricinus occurs throughout a large part of continental Europe and the species is assumed to be nearly omnipresent in Germany (Estrada-Peña, Mihalca & Petney, 2018; ECDC, 2019).

Transmission cycle

Borrelia burgdorferi bacteria and the TBE viruses circulate between I. ricinus individuals and their vertebrate hosts. Ticks become infected when they feed on birds or mammals that carry the pathogens in their blood and may transmit the pathogens with the following blood meal to the next host (Stanek et al., 2012). Adult ticks usually feed on larger animals such as deer, which are not a reservoir for B. burgdorferi bacteria, but help to maintain the ticks’ reproductive stage (ECDC, 2020). The main reservoir hosts of TBE virus are mainly small rodents, e.g., voles and mice, but also insectivores and carnivores (CDC, 2020; ECDC, 2020). Ticks, specifically hard ticks of the family Ixodidae, act as both the vector and reservoir for TBE viruses (CDC, 2020).

Transmission of B. burgdorferi to humans occurs through a bite from an infected tick, in most cases by a tick in the nymph stage (Gray, 1998). Being more numerous than adult ticks, nymphal ticks are responsible for approximately 80% of tick bites in many areas (Heyman et al., 2010) and, thus, we may expect them to cause most of LB cases. Larval tick bites do not pose a significant risk because larvae rarely carry infection (Gray et al., 1998). In field collections less than 1% of host seeking larvae have been found to be infected with B. burgdorferi (Hofhuis et al., 2017) which may be attributed to transovarial transmission (Humair & Gern, 2000).

Infected ticks are unlikely to transmit the B. burgdorferi s.l. in the early hours of a feed, but the risk rises steadily the longer the blood meal takes; so early removal of attached ticks within the first hours is very useful in reducing transmission risk (ECDC, 2014).

In contrast, the TBE virus can even be transmitted from the saliva of an infected tick within the first minutes of the bite (Lindquist & Vapalahti, 2008).

Factors affecting the number of human tick-borne disease cases

The risk of human infection by TBDs depends on many environmental factors and their interactions, affecting both the ecology of the pathogens and that of the vectors and reservoir hosts (Süss, 2003). The number of TBD cases in humans thus primarily depends on the abundance of tick species, the frequency of contact between humans and ticks and the prevalence of the pathogens. The higher the number of ticks, the higher the risk for humans to be bitten; the higher the prevalence of pathogens, the higher the risk of becoming infected through a tick bite (Randolph, 2008; Pfäffle et al., 2013; Wikel, 2018; Escobar, 2020).

The abundance of ticks depends on climatic and weather conditions (e.g., mild winters favour a high abundance in the following season), land cover (e.g., higher abundances in forested areas) and the abundance of hosts (e.g., small rodents, birds, red deer and wild boars) (Randolph, 2008; Jaenson & Lindgren, 2011; Jaenson et al., 2012; Medlock et al., 2013). Temperature can also have a direct impact on the transmission risk of pathogens as it influences reproduction of pathogens in ectothermal vector species. Thus, it is assumed that higher temperatures have a direct effect on the risk of infection with tick-borne diseases (Samuel, Adelman & Myles, 2016).

There are several interactions between the components that influence the risk for humans of becoming infected with tick-borne pathogens (Fig. 1). The prevalence of TBD agents in ticks also depends on the availability of hosts and the prevalence of TBD agents in hosts. In particular, it is assumed that the abundance of small rodents depends on the availability of food (depending on the fructification of forest trees) and weather conditions (Klempa, 2009; Reil et al., 2015). In spatial terms, land cover may play an important role (e.g., forest cover) (Randolph, 2010; Pfäffle et al., 2013). Human behaviour depends on the weather situation and on land cover (Randolph, 2010): People spend more time outside in good weather than in bad weather and forested regions may be more crowded than other land cover types. For TBE, vaccination coverage is also an important factor (Randolph, 2008).

Figure 1 Factors that potentially affect the number of recorded human tick-borne disease cases.

Aims and scopes

In this study, we display the spatial and temporal (annual and seasonal) patterns of LB and TBE based on the data provided by the Robert-Koch-Institute (RKI) between 2001 and 2019. We discuss and compare these patterns considering possible underlying drivers: climatic and weather conditions and other factors (see Fig. 1).

Materials and Methods

We tried to link additional data with the spatial and temporal patterns of the reported cases of the two tick-borne diseases in Germany. All maps are based on data downloaded from various databases, and data covered the following variables: Data on reported cases of LB and TBE (from hereon after “reported cases” refers to human cases only) was downloaded at a spatial level of German administrative districts (German: “Landkreise”) for the years 2001 to 2019 with a temporal resolution of one calendar week (Robert Koch-Institut: SurvStat@RKI 2.0, https://survstat.rki.de, query date: 03.02.2020) and mapped using the geometry of NUTS250 (Nomenclature des unités territoriales statistiques) data provided by the German Federal Agency for Cartography und Geodesy (GeoBasis-DE/BKG (2020)) and using Esri ArcGIS software (Esri, Rednands, CA, USA).

Data on population numbers in each of the German administrative districts were taken from the German Federal Agency for Cartography und Geodesy (GeoBasis-DE/BKG (2020)-status of the documentation 01.12.2017, accessed on 02.03.2020).

Occurrence records for Ixodes ricinus as the major vector of TBE and LB were taken from the Global Biodiversity Information Facility (GBIF) data base (www.GBIF.org, DOI 10.15468/dl.yak5vd (GBIF), accessed on 23.06.2020). We also used information of the currently known distribution of I. ricinus by administrative districts based on data from the European Centre for Disease Prevention and Control (ECDC, 2019).

Taking the wild boar (Sus scrofa) as one example of an important host species, we used data on its spatial distribution at the level of administrative districts (Nuts3 units), more specifically, the number of reported individuals (hunted and roadkill) per 100 ha of hunting area in the respective administrative district. The data was compiled by the German Hunting Association (Deutscher Jagdverband DJV) as part of the WILD monitoring (DJV, 2020). We chose the wild boar as a representative of a large host of ticks because of data available and because we assume that other medium and large forest-associated hosts (roe deer, red deer, red fox…) of ticks show similar ecological requirements and therefore similar distribution patterns (see also below-h).

Data on temperature and precipitations patterns in Germany was taken from the German Meteorological Service (Deutscher Wetterdienst DWD). We considered the mean annual temperature and precipitation, both averaged over the empirical data from 1981 to 2010 per grid cell (1 km × 1 km) for spatial patterns, and the mean monthly mean air temperature and precipitation from 2001 to 2019 for temporal patterns.

The percentage of forest area in each German administrative district was derived from the Corine landcover data (CLC, 2012) provided by the Copernicus Land Monitoring Service, coordinated by the European Environment Agency (EEA) and includes deciduous forest, coniferous forest and mixed forest (3.1.1, 3.1.2, 3.1.3).

Data on beech mast stem from reports on the state of the forests (in German: “Waldzustandberichte”) of individual German federal states. Since the data are not available for all federal states over the entire period, the values of the federal states Baden-Wuerttemberg (WZB, 2018a), Bavaria (WZB, 2019a), Brandenburg (WZB, 2018b), Hesse (WZB, 2019b) and Saxony (WZB, 2019c) were averaged for each year between 2002 and 2019. Despite observed smaller regional differences in beech mast (Reil et al., 2015), it is generally assumed that mast years are largely driven by weather conditions that have occurred in previous years in combination with refractory periods to accumulate resources (Piovesan & Adams, 2001; Overgaard, Gemmel & Karlsson, 2007; Cunze et al., 2018). We therefore argue that the averages of the empirical data from the above mentioned five federal states are a good measure for the average situation in Germany.

Occurrence data for selected small rodents (main hosts for ticks in larval and nymph stages), medium and large mammals (main hosts for adult ticks and important dispersal vectors for ticks) as well as bird species (also serving as hosts and dispersal vectors for ticks) were taken from GBIF (www.GBIF.org, GBIF, 2021).

A map including names of federal states and single administrative districts is given in the appendix (Fig. S1). All maps were generated using Esri ArcMap, version 10.8.1.

Results

Spatial patterns of reported TBE and LB disease cases in Germany 2001-2019

The spatial distribution of the incidence (reported cases per 100,000 inhabitants) of TBE in Germany (Fig. 2) shows a very similar pattern for all years between 2001 and 2019, with a larger number of cases in the south of Germany and far fewer cases in the north. In the north, no cases have been reported in the majority of German administrative district, whereas up to 13 cases per 100,000 inhabitants have been reported from administrative districts in the south, with the highest number of cases located in the federal states Bavaria and Baden-Wuerttemberg, South Hesse, South-East Thuringia and Saxony. Generally, the number of recorded TBE cases seems to increase over time and individual districts seem to be particularly affected but without a temporal pattern.

Figure 2 Spatial patterns of incidences (reported cases per 100,000 inhabitants) for tick-borne encephalitis (TBE) at the level of German administrative districts (RKI data).

The spatial pattern of LB infections over time in Germany is depicted in Fig. 3. In many federal states, the obligation to report LB infections was first introduced in the period between 2001 and 2019 (e.g., Saarland and Rhineland-Palatinate in 2011 and Bavaria in 2016) or is still not in place (e.g., Baden-Württemberg, Hesse, North Rhine-Westphalia, Lower Saxony and Schleswig-Holstein) and explains the large amount of grey area with no available data on LB cases. Due to the different regulations on reporting obligations in the federal states of Germany, spatial patterns are difficult to assess. Considering areas with existing reporting obligations, there are no clear patterns such as gradients or hotspots.

Figure 3 Spatial patterns of incidences (reported cases per 100,000 inhabitants) for Lyme borreliosis (LB) at the level of German administrative districts (RKI data).

With regard to factors assumed to influence the spatial patterns of tick-borne disease occurrence, we have mapped spatial patterns in the occurrence of each component in the transmission cycle (humans, Ixodes ricinus as the main vector species and wild boar density, representing the occurrence of hosts) and several relevant environmental factors (precipitation, temperature and forest cover).

Figure 4 may help to interpret the spatial patterns of the human infections with TBE and LB. Since there is no nationwide reporting obligation for LB in Germany, we have only shown the mean data for the six federal states in which there has been a continuous reporting obligation since 2002 in Fig. 4B.

Figure 4 Spatial patterns of tick-borne diseases, and associated environmental factors.

(A) Sum of recorded human TBE cases at a spatial level of German administrative districts between 2002 and 2019 (RKI data); (B) Sum of recorded human LB cases at a spatial level of German administrative districts between 2002 and 2019 (RKI data); (C) number of inhabitants (RKI data). (D) occurrence records for Ixodes ricinus (GBIF data) and areas with known presence of I. ricinus (ECDC); (E) distribution of wild boar (WILD data DJV); (F) annual precipitation [mm] (DWD data); (G) annual mean temperature [°C] (DWD data); (H) percentage forest cover (Corine land cover data).

For example, the district Sächsische Schweiz Osterzgebirge in the south of Saxony (cf. Fig. S1), which has a high total number of reported LB cases in the period 2002 to 2019 (4,116 cases, Fig. 4B), is characterized by a relatively high population density (Fig. 4C), average wild boar density (Fig. 4E) and rather low forest cover (Fig. 4H). Temperatures are also comparatively low with a lot of precipitation (Figs. 4F, 4G). On the other hand, the hotspot for LB in 2016 and 2018 for LB in eastern Bavaria (Fig. 3) in the administrative districts Regen and Freyung-Gravenau (cf. Fig. S1). These two districts are characterized by a low population density (Fig. 4C), a high proportion of forest (Fig. 4H) and a relatively low density of wild boar (Fig. 4E). Ticks were reported from these areas (Fig. 4D). The temperatures are comparatively low with high precipitation (Figs. 4F, 4G).

The North-South gradient in the number of human TBE infections in Germany (Fig. 2) is not reflected by any of the factors displayed in Figs. 4C–4H.

In the temporal patterns, the numbers of reported cases of TBE (Fig. 5A) and LB (Fig. 5A) show seasonal variations with low numbers of cases in the winter months. It can be seen, for example, that 2012 with remarkably low numbers of recorded cases for both diseases (Figs. 5A–5D) was preceded by four winters with very low temperatures in winter (Fig. 5E). The year 2018 with comparably high numbers of recorded cases for both diseases (Figs. 5A–5D) was preceded by two cold winters (Fig. 5E) but also a strong beech mast year in 2017 (Fig. 5G). Overall, the temperature conditions cannot be clearly related to the temporal patterns in the reported TBD cases. For precipitation (Fig. 5F) it is also difficult to link the temporal patterns.

Figure 5 Temporal patterns of the recorded human tick-borne diseases cases between 2002 and 2019 in relation to the main components in the transmission cycle and associated environmental factors.

(A) Sum of recorded human TBE cases in Germany per month (RKI data); (B) Sum of recorded human LB cases in the six federal states with a continuous reporting obligation since 2002 (Brandenburg, Berlin, Mecklenburg-Vorpommern, Saxony, Saxony-Anhalt and Thuringia) per month (RKI data); (C) Recorded human TBE cases in Germany per year (RKI data); (D) Recorded human LB cases in the above mentioned six federal states per year (RKI data) (E) monthly mean temperature [°C] (DWD data); (F) monthly precipitation sums [mm] (DWD data); (4G) beech mast: percentage of trees with strong fructification mean over data of five federal states (WZB, 2018a, 2018b, 2019a, 2019c, 2019b).

In the period under consideration, 2002 to 2019, no significant trend could be observed in the annually reported case numbers for TBE and a slightly positive, significant trend (p-value = 0.02402) in the reported LB numbers (all cases reported in the six eastern German federal states: Brandenburg, Berlin, Mecklenburg-Vorpommern, Saxony, Saxony-Anhalt and Thuringia).

However, there are strong fluctuations between years. The annual fluctuations of both diseases move in parallel. A clear correlation of disease cases of TBE and LB can be seen when looking at annual and monthly resolution (Fig. 6). Years with a high number of TBE cases also show high numbers in LB cases and vice versa. When looking at the monthly data, an even stronger correlation becomes apparent, which is partly due to the similar seasonal patterns.

Figure 6 Relation between numbers of reported disease cases of TBE and LB between 2002 and 2019.

Scatterplot of the reported cases reported in Germany for TBE and in the six federal states with a continuous reporting obligation since 2002 (Brandenburg, Berlin, Mecklenburg-Vorpommern, Saxony, Saxony-Anhalt and Thuringia) for LB (to ensure comparability of the data) with a temporal resolution of (A) 1 year and (B) 1 month, respectively. Pearson correlation coefficient r = 0.71 (p < 0.001) for the annual data and r = 0.85 (p < 0.001) for the monthly data. RKI data.

The gender distribution of both TBDs cases, shows consistent patterns across different age groups for LB (Fig. 7A): with higher incidences for females in the age categories 30-39, 40-49, 50-59, 60-60 and 70-79 years and for TBE (Fig. 7B): higher incidences for males in all age groups.

Figure 7 Numbers of recorded TBD cases related to gender and age.

(A) Numbers of reported LB cases; (B) Numbers of reported TBE cases (RKI data).

Discussion

Ticks are important disease-transmitting vectors. Vector-borne diseases are on the rise in Europe, which has been attributed to climate-change induced range shift of the tick vectors. Here we described the spatial and temporal patterns of human TBE and LB cases in Germany in relation to possible underlying drivers. There are many factors assumed to affect the spatial-temporal patterns of human tick-borne diseases and their interrelationships are complex.

We found the temporal patterns to fluctuate strongly between years, and the patterns for both diseases are relatively synchronized. The strong connection between the numbers of cases of both diseases in the temporal patterns suggest that similar driving forces are involved. The presence of I. ricinus as the main vector of both diseases in Europe is the key determinant for the incidence of both diseases.

The temporal and spatial patterns presented in this study are considered typical for both diseases and are in accordance with those shown in studies in other European countries (e.g., Daniel et al., 2008; Randolph, 2010; Moore et al., 2014). The occurrence of both diseases is highly seasonal. The seasonal patterns with a peak in summer and a subsequent decline to nearly zero in the winter months in the reported TBE cases and only to a few LB cases are clearly attributed to the annual cycle of the ticks with a diapause in winter and questing behaviour during their activity phase. This is obvious, as the diseases cannot be transmitted without blood-sucking ticks (apart from a small number of TBE cases associated with the consumption of raw milk). In addition, small mammals that act as hosts are also almost inactive during the winter months.

The seasonal patterns of reported cases of both diseases are very similar with a slightly earlier peak for TBE whose German name translates to “Early Summer Meningoencephalitis”. Randolph (2010) reports that the seasonal patterns in the activity of questing nymphs show a similar pattern, but with peaks spared again towards earlier times of the year. This could generally be explained by the incubation time and delay in reporting (Robert Koch-Institut, 2007).

The fact that the number of reported LB cases does not reduce to zero in winter despite the assumed inactivity of the transmitting ticks can be attributed on the one hand by a delayed diagnosis. On the other hand, increased tick activity during the winter months is reported (Dautel et al., 2008), which may be due to changed climatic conditions, especially milder winter temperatures in the northern hemisphere (Gray et al., 2009). It is also expected that warmer and drier summers could also affect the seasonal patterns in tick activity and hence the seasonal patterns in the occurrence of the associated diseases (Gray et al., 2009).

There is a high variation between years, in which the tick-borne diseases occur more frequently and years with comparatively low numbers of reported cases. It is of great interest to understand the underlying factors driving this fluctuation and to be able to estimate in advance whether high numbers of cases are to be expected in the current year.

There is no doubt that climatic and weather conditions impact tick activity and thus the temporal patterns of tick-borne diseases (Moore et al., 2014). Ticks are known to be very sensitive to temperature and humidity, their requirements and sensitivities differ between the stages of the life cycle which leads to temperature and humidity conditions influencing tick population dynamics, demographic processes and phenology regarding activity and questing behaviour (Moore et al., 2014). Development rates of ticks are favoured by warmer temperature. Cold winter temperatures and persistent drought in summer are considered to be stress factors for ticks and can lead to reduced abundances in years following a cold winter and to reduced activity during dry summers (Gray et al., 2009). In this context, Bennet, Halling & Berglund (2006) reported an increased incidence of LB cases after mild winters and during humid and warm summers in southern Sweden. For Germany, we did not find evidence for a clear correlation between the number of human TBD cases and climatic or weather conditions, which is probably due to the complexity of all underlying associations and factors. Besides the obvious impact of weather and climatic conditions, there are important non-entomologic factors impacting the occurrence of tick-borne disease in humans. So, human behaviour, which may favour the contact rates between questing ticks and humans, is regarded as a key factor. Here, climatic or weather factors can also play an indirect role, as favourable weather conditions promote outdoor activities such as mushroom picking (Daniel et al., 2008). Outdoor activities also depend on spatial factors such as forested area or attractive, touristically developed areas.

We found a clear pattern that LB tends to be diagnosed more often in women than in men (Fig. 7A). TBE, on the other hand, is more common in men than in women (Fig. 7B). Similar patterns can be found in other countries (e.g., Zöldi et al. (2013) in Hungary and Sun et al. (2017) for TBE in China) but also different patterns (e.g., Bacon, Kugeler & Mead (2008) for the United States).

In the case of TBE, the vaccination coverage has an important impact. Vaccination is recommended for people in risk areas and for people who are exposed to an increased risk due to their activities (e.g., forest workers). Vaccination coverage varies strongly between regions (Robert Koch-Institut, 2020) and ranged in TBE risk areas in the German Federal States Bavaria, Baden Wurttemberg, Hesse and Thuringia from 20–41% for children at school entry (Hellenbrand et al., 2019). It is assumed that the proportion of vaccinated children is higher than the proportion of vaccinated adults (>15 years Robert Koch-Institut, 2020) but data are currently only available for children at school entry (Hellenbrand et al., 2019).

The significant slightly positive trend in TBE case numbers found by Hellenbrand et al. (2019) for the period 2001 to 2018 could not be confirmed in our study for the period 2002 to 2019, which could be due to the slight decrease in case numbers for 2019. However, to derive reliable evidence for the promotion of tick-borne diseases in Germany due to climate change, longer time series would be required, as well as a standardised reporting and surveillance system. Regarding the number of reported LB cases, a slightly positive trend can be seen. However, this can be attributed in particular to the increase in the period 2002 to 2006, a period in which the reported numbers are subject to additional uncertainties typical of the initial phase of reporting practice. In general, it can be assumed that the number of reported LB cases significantly underestimates the actual number of cases (Robert Koch-Institut, 2007; Palmieri et al., 2013). After an initial phase of a surveillance program, the awareness of the population and the diagnostic experience of the medical staff increases over time (Robert Koch-Institut, 2007).

With regard to the spatial but also to the temporal patterns of tick-borne diseases, distribution patterns of host species are of major importance. The most important reservoirs are mice and birds. Voles seem to play an important role as reservoir hosts for TBE, as the virus appears to be well adapted to voles (Zöldi et al., 2015). In addition, other animals such as reptiles, hedgehogs, foxes or rabbits are also regarded as host animals and pathogen reservoirs. In this context, the fructification of forest trees can be considered as a potential driver for temporal fluctuations in the abundance of hosts, ticks and consequently the incidences of TBE and LB. A high amount of fruit means a good nutritional basis and thus higher abundance for host species such as roe deer, red deer, wild boar and mice. Ostfeld, Jones & Wolff (1996) reported an increase in nymph abundance of Ixodes scapularis 2 years after heavy fructification in North America. Distributional patterns of larger host species like dears are also affected by alternating grazing supply, hunting pressure and, in Germany increasingly, by wolf occurrence. Tick abundance has been positively associated with deer abundance (Gilbert, 2010b). We exemplarily displayed occurrence records of three small tick hosts (Myodes glareolus, Apodemus flavicollis and shrews of the Soricidae family, Fig. S2), larger tick hosts (Capreolus capreolus, Cervus elaphus and Vulpes vulpes, Fig. S3) and migratory birds (Grus grus, Ciconia ciconia and Turdus merula, Fig. S4). A pattern of these species’ distributions that explains the spatial patterns in tick-associated diseases does not seem obvious. The distribution pattern of the mammals may be similar to the distribution of the forest areas (Fig. 4H). However, all the maps with observed occurrences provided by GBIF of the here considered species slightly show a coincident pattern with a clustering of points e.g., along the Rhine river (south-western Germany). This could also indicate a certain sampling bias, that in these areas the number of reports is generally higher.

Regarding spatial dispersal patterns, birds and particularly migratory birds may be of particular importance. Migratory birds have been found to carry not only ticks but also TBD infected ticks (Hasle et al., 2009; Hasle, 2013). They thus not only serve as hosts for ticks and reservoir hosts for TBD pathogens (Wilhelmsson et al., 2020) but can also play an important role as potential long-distance dispersal vectors for ticks and pathogens. Attached to birds, ticks and associated pathogens can cover large distances, cross geographic barriers (e.g., rivers, oceans and mountains), and possibly be introduced into new areas (Hasle, 2013). Detailed data on resting places of migratory birds in Germany are poorly available. However, for example the regions rich in lakes in eastern Germany are considered as breeding places for e.g., cranes. No increased incidence of TBD cases could be detected in these areas, nor could clear correlations between TBD distribution and the distribution patterns of selected bird species (Figs. S2–S4) be identified. In Germany, the blackbirds (Turdus merula) are considered to be one of the most important species for harbouring ticks (Klaus et al., 2016). This species is very common and widespread in Germany.

To conclude, the interactions and interrelationships are complex. For further investigation of the underlying driving factors and their interrelationships to achieve better risk assessment, longer time series and standardized reporting and monitoring programs would be valuable.

Supplemental Information

Supplemental Information 1 Federal states of Germany and certain administrative districts mentioned in the text.

Click here for additional data file.

Additional Information and Declarations

Competing Interests

Author Contributions

Data Availability

The authors declare that they have no competing interests.

Sarah Cunze conceived and designed the experiments, analyzed the data, prepared figures and/or tables, authored or reviewed drafts of the paper, and approved the final draft.

Gustav Glock analyzed the data, prepared figures and/or tables, authored or reviewed drafts of the paper, and approved the final draft.

Sven Klimpel conceived and designed the experiments, authored or reviewed drafts of the paper, and approved the final draft.

The following information was supplied regarding data availability:

All data used in this study is freely available:

• Data on temperature and precipitation conditions were obtained from the Climate Data Center (CDC) of Deutscher Wetterdienst (DWD) (https://opendata.dwd.de), under the terms of the GeoNutzV 2013 license: https://www.dwd.de/EN/service/copyright/copyright_node.html.

• NUTS https://gdz.bkg.bund.de/index.php/default/nuts-gebiete-1-250-000-stand-31-12-nuts250-31-12.html.

• CORINE land cover: https://gdz.bkg.bund.de/index.php/default/corine-land-cover-5-ha-stand-2012-clc5-2012.html.

Both are licensed under by the Data licence Germany–attribution–version 2.0, http://www.govdata.de/dl-de/by-2-0.

• GBIF occurrence data:

Apodemus flavicollis: https://doi.org/10.15468/dl.p4r693 (01 June 2021),

Capreolus capreolus: https://doi.org/10.15468/dl.wp98ts (28 May 2021),

Cervus elaphus: https://doi.org/10.15468/dl.72ddmz (28 May 2021),

Ciconia ciconia: https://doi.org/10.15468/dl.k6jv69 (28 May 2021),

Soricidae https://doi.org/10.15468/dl.czyyq2 (01 June 2021),

Grus grus: https://doi.org/10.15468/dl.j4fe7k (28 May 2021),

Ixodes ricinus: https://doi.org/10.15468/dl.yak5vd (23 June 2020),

Myodes glareolus: https://doi.org/10.15468/dl.82wsgb (01 June 2021),

Turdus merula: https://doi.org/10.15468/dl.t3h8b4 (28 May 2021),

Vulpes vulpes: https://doi.org/10.15468/dl.wgtneb (28 May 2021).

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
