# Peer review of "Spatial and temporal distribution patterns of tick-borne diseases (Tick-borne Encephalitis and Lyme Borreliosis) in Germany"

_PeerJ, doi:10.7717/peerj.12422_

## Round 0.1 · original submission · Major Revisions

The review process is now complete, and two thorough reviews from highly qualified referees are included at the bottom of this letter. Two reviewers including myself agree the manuscript deserves to be published. Although there is considerable merit in your paper, we also identified some concerns that must be considered in your resubmission. Please, consider addressing the limitations in relation to the findings of the study accordingly to reviewers’ comments and suggestions.

·

Basic reporting

Cunze et al. gathered spatial and temporal patterns of reported cases of
human TBE and LB and some associated factors, in order to find some association between these factors and curve of human TBD cases. Figures, the language, editing are good.
The authors overtook a heavy burden as lot of researhers tried to explain epidemiological data of TBDiseases by some associated factors, but they could not. This is also the case with this paper. Although the manuscript is valuable and worth publishing, but the aimed goal is unaccessable, as TBDs are under preassure of very complex effects of lot of known and still unknown factors.

Experimental design

Other factors (more game animals, migratory birds) could have been included in the study, see later. Otherwise the plan, figures, are Ok.

Validity of the findings

The authors give interesting details of the LB, TBEV epidemiological processes of the last decade in Germany, which are valuable for researchers, but real findings i.e. association of some of the studied factors to LB TBEV epidemic data I think is missing. As such association have never been done by others so far.

Additional comments

Introduction.
Reading the introduction I could not decide whether this paper was planned as an informative review article for the public, or for collegues from the TBD scientific field. The introduction is full of data which are known and obvious for researchers. When we are citing we must cite those authors which described first the cited data. e.g. bimodal pattern of tick activity was described decades earlier than Estrada-Pena et al. (otherwise one of our ongoing study of ticks of hedgehogs detected 5-6 small peaks for all tick stages, not 1 and not 2). The thing is not so simple as bimodal or unimodal. I would suggest the authors review papers of Mandl et al., 2008, Süss 2003 Vaccine 21, 2011, Gray 1991. (and several others).

Reference Mehlhorn H. 2012. For me, who does not speak German such citations are not understandable, unaccessable, and as such, does not exists. Cite papers avaiable online with minimum an English abstract. Especially when the cited things were described earlier by others, cite the previous author(s).

row 84 Lucius et al., article is missing from the reference chapter

row. 112. I am a little bit doubtful, whether an adult female could spent minimum 2-3 days on a human being to infect him/her with LB spirochaete, whitout being noticed. Human LB infections must be made mostly by nymphs, whose bloodmeal is much faster, and could be remained unobserved throughout their biting. The same for TBE with larvae.
115-116. Larvae can transmit TBEV, even in the first minutes of biting. Could have been mentioned here.
120-121- „plus human beings nearby” is also necessary
Figure 3. It is not easy to evaluate epidemiological processes with lack of 2/3, or half of the necessary data.
254-256- The two illnesses are parallelly move. Should be mentioned underlined, and think of it.
260-261- why the two brackets are necessary here?
267- small mammal host species are also almost inactives.
303- …. is repeatedly suspected, „but still not proven” .
312--313 With both disease subclinical infection exist, especially with TBEV. The reported data is only the number of infections with symptoms. + problem of standardized reporting, different laboratory (detection methods) background make most of our epidemiological data not reliable. We could draw other conclusions if we knew the real number of infected humans.
323-324.- Red voles seems the most important small mammal vector especially for TBEV. Please see papers below, and there are some others, among them articles from German authors.
331 Oh dear, not dear but probably deer
334-335 it is not true. see previously
335- co-feeding. On the base of a badly written, unnatural, laboratory, forced paper of Labuda et al, 1993 Vet. Med entomol. (the worst paper I have ever read) there is no proof for co-feeding process in nature in case of TBEV or LB transmission (and probably it is also true for other TB agents). This fairy tale successfully were smuggled into hundreds of brains, and researchers (who never did anything with co-feeding just read about it) cite those who cited others. Finally this co-feeding idea came into fashion, and accepted without any proof. No independent author studied the phenomenon in nature, so the whole theory is still a theory. Personally after capturing hundreds of small mammals in nature, and sampled them for ticks, no co-feeding ticks were found, or seen. Some authors (S Randolph ) even writes, that biting viremic hosts is negligable in nature comparing to the role of the miraculous co-feeding (her enduring paradigm). We should forget this co-feeding, till somebody, independent authors really prove it. Please mention this co-feeding if You have any proof that this theory exists at all.
340-342- competent not competent. Large mammals like red deers are able to move 60 km a day. By transporting engorging females deers are able to create new foci, at areas where noone expects ticks, or tick-borne pathogens. Role of large mamals in TBdiseases is not only feeding females but to spread them.

General comments:
What I missed.
migratory birds. What about the role of migrating birds in TBD foci in Germany. It has been shown that migratory birds carry not only ticks, but TBD infected ticks. REF Hasle, Wanderström. Recently TBEV strains (with almost identical genome sequences) were isolated in Bayern (XY), Finland (XY) and in Hungary (Egyed et al., 2018). All the three isolation sites were waterfowl habitats, and probably bound by migrating birds, as Hungary is the main resting place of Central-East-European migrating birds crossing the Bosporus. Are there places in Germany where large body sized migrating birds (storks, cranes, Anseriformes) accumulate, rest? How these places are related to the human TBD cases?

females/males. What about the sex/gender of human patients in Germany? In a similar brother-paper from Hungary (which should have been read and cited) 70% male dominance was found among TBEV patients, but a slight female majority 54% among LB patients. Is this similar in Germany, or not? For both, what could be the reason?
Zöldi, V., Juhász, A., Nagy, Cs., Papp, Z., Egyed. L. 2013. Tick-borne encephalitis and Lyme disease in Hungary: the epidemiological situation between 1998 and 2008. Vector borne and zoonotic diseases. 13(4):256-265. doi:10.1089/vbz.2011.0905

why wild bores? Selecting only one (2) species could be misleading. Wild ruminants especially roe deers probably are better hosts for ticks than wild pigs. Roe deer live in a relatively small area (in contrary to boars) lies a lot on the ground regularly, daily, in the grass or in remote bushy areas. In my country we have 1.2 boars /km2, but 4 roe deers/km2. From your country Król et al., 2020 Parsitol res) roe deer was also found as good target for tick infestation, Haut et al, 2020 Microroganisms) found correlation from German red foxes and human TBEV cases. An Italian work also found corelation between roe deer abundance and human TBEV cases but these case swere not corelated with any climate factors tested (Rizzoli et al., 2009, Plos One). Swedish authors found positive correlation between TBE human cases and population densities of roe and red deers and hares, but not with fallow deer, elks, red fox, lynx and wild boar. So densities of certain game animals are correlate with human TBD cases, others not. Similarly, more (or ruminant) wild species should have been studied instead of wild boars. It would have been interesting to compare German and Swedish data.
Jaenson, GTT., Petersson, H.E., Jaenson, G.E.D., Kindberg, J., Pettersson, H-O.J., Hjertqvist, M., Jolyon M Medlock, M.J., Bengtsson, H., 2018. The importance of wildlife in the ecology and epidemiology of the TBE virus in Sweden: incidence of human TBE correlates with abundance of deer and hares. Parasit Vectors 11(1):477. doi: 10.1186/s13071-018-3057-4.

What about vaccination (TBEV) of German people? What percentage of them are vaccinated and in which provinces (states)? Vaccination of the population against TBEV is a decisive tool to decrease TBEV human cases, and to change completely the epidemiological situation. In two European countries (quite close to Yours) vaccination almost annihilated the TBEV epidemics. In my country the 3 hospitalized TBEV patients/100000 inhabitants after 1995-96 was sharply dropped to 0.2-0.6 due to a large-scale vaccination champaign.

Figure 3. It is very difficult to draw reliable conclusions, when half, 2/3rd of the neccesary data are missing. If all provinces of Germany would release their LB data some time in the future this work should have been repetead probably with better results.

Climate change 126-128- in our published papers (I not cite them) we never found correlation between abundance, activity of ticks, and any weather conditions (min. max. mean daily temperatures, rain humidity etc.). Jaenson et al + Rizzoli, and lot of others (above) also could not. Someone can assume lot of things.
Like with co-feeding authors should be more careful with this climate chance-tick abundance moving Northward idea. No black and white proof for that is exists. During an EU project (2005-10) dozens of TBE researchers from Europe believed it, and tried to prove it. Finally the project leader must admitted, that altered human behavioural habits, leisure-time activities (mushroom picking) is much probable reason for changed human TBEV epidemiological data, the increased human cases than climate change (Sarah Randolph 2010, You cited). Even Your own data and sentences, rows 232-233, 236-238 (cold does not has effect on human TBD cases) rows- 296-298 298-301 (warm also not) admit, and what is assumed, does not work. If climate change (warming) would increase human TBEV cases or tick abundance, moving northward etc. in the Nothern half of Germany TBEV would not be almost unknown. (Your Figure 2).
Even the references You cited (Semenza and Suk, Medlock) are full of assumptions. There are no proven effect of climate change onto the objects of this manuscript (TBEV, LB, ticks ). There is no need to wash together things which are not belong together (mosquitoes and ticks, West Nile/chikungunya and TBEV/LB).
The authors should indicate both in Abstract and in the Discussion chapter, that correlation between human TBEV, LB human cases and any of the tested climate factors were not established.

VOLES

Zöldi,V., Papp, T., Reiczigel, J., Egyed L. 2015. Bank voles and adult rodents show high seropositivity rates in a natural TBEV focus in Hungary. Infectious diseases. Mar 2015, Vol. 47, No. 3: 178–181.

Michelitsch A, Tews BA, Klaus C, Bestehorn-Willmann M, Dobler G, Beer M, Wernike K. In Vivo Characterization of Tick-Borne Encephalitis Virus in Bank Voles (Myodes glareolus). Viruses. 2019 Nov 15;11(11):1069. doi: 10.3390/v11111069.

Grzybek M, Tołkacz K, Alsarraf M, Dwużnik D, Szczepaniak K, Tomczuk K, Biernat B, Behnke JM, Bajer A. Seroprevalence of Tick-Borne Encephalitis Virus in Three Species of Voles (Microtus spp.) in Poland. J Wildl Dis. 2020 Apr;56(2):492-494. doi: 10.7589/2019-02-048. Epub 2019 Dec 27.


Tonteri E, Kipar A, Voutilainen L, Vene S, Vaheri A, Vapalahti O, Lundkvist Å. The three subtypes of tick-borne encephalitis virus induce encephalitis in a natural host, the bank vole (Myodes glareolus). PLoS One. 2013 Dec 13;8(12):e81214. doi: 10.1371/journal.pone.0081214. eCollection 2013

Reviewer 2 ·

Basic reporting

The manuscript describe a well conducted and informative analysis of the spatial and temporal variation of two diseases, Tick-Borne Encephalitis (TBE) and Lyme Borreliosis (LB), in Germany and some possible underling drivers. The manuscript is well written and clear, and figures are nice. The authors provide the necessary background for the study, so that, even non-specialists in the system can understand it.

The only important issue, but easy to solve, is that some statements that are assumptions of the study are provided in the introduction without proper references. See this and other minor suggestions bellow.

Experimental design

The study is well designed, the research question is well defined and relevant.
As far as I can tell, the data is appropriate for the question addressed, the analysis is well conducted, resulting in meaningful results.

Validity of the findings

My main criticism to the manuscript is that the discussion is a bit disconnected from the results presented.

The study did not find a relationship between climate and other factors and the temporal variation of the diseases cases. But most of the discussion is written as if the pattern was detected. Because of this disconnection, the reader might feel that the absence of pattern is an unwelcome finding. And that is unfair to the good study described in the manuscript.

The authors should, of course, present other perspectives, acknowledge the limitations of their study and try to explain why the results did not match the expectations. But the discussion must be mostly focused on what they found, not on what they expected to find.

Additional comments

Specific suggestions:

Line 35 – As several mild winters did not resulted in increased case numbers and cold winters did not resulted in decreased case number, linking the low case numbers of 2012 with the previous cold winters is a too unsupported speculation.

Line 42 – The conclusion of the study should be that, given the analyzed evidence, weather conditions did not influence the diseases’ prevalence.

Line 52 – Probably a typo here.

Line 112 – I suggest reformulating the sentence to separate what is known, from what is the possible consequence, for instance “Being more numerous than adult ticks, nymphal ticks are responsible for approximately 80% of tick bites in many areas (Heyman et al., 2010) and, thus, we may expect them to cause most of LB cases”.

Line 114 -115 - These are too important information to be provided without references.

Line 122-128 – These are the main assumptions that justify the study. Authors should provide references for them.

Line 130 – The prevalence of the pathogens depends on the prevalence of the pathogens (?)

Line 132 – It is assumed by whom? If this is a common approach, the authors should provide references. Otherwise, the author should state “Here, we assume …” and justify why their assumption is reasonable.

Introduction - The introduction is very well written and is accessible for non-specialist in LB and TBE diseases, and even, for non-parasitologists. All the important information is provided. An easily solvable but important weakness is that essential assumptions for the study are provided without references. Also, I strongly suggest the authors to include a last paragraph on the introduction explicitly describing the aim of the study and briefly explaining the study design. This paragraph will help readers to interpret the material and methods and benefit the flow of reading. Such a section is present in most of PeerJ recently published papers.

Line 138 – An improved version of this paragraph could be the last introduction paragraph.

Line 206 – What do the authors mean by that? Are there different regulations on reporting between federal states on the North and on the South of Germany? Do they suspect that these regulations are the causes for more reported cases in the south? Otherwise, I strongly disagree with this sentence. The spatial pattern (South vs North) is absolutely clear.
Maybe what the authors mean is that it is difficult to assess the spatial pattern in a smaller geographical scale (e.g., among administrative districts within each Bundesland). Anyway, this sentence is unnecessarily undermining their findings.

Line 213 – Here, for LB, the sentence in line 206 make sense.

Line 215 – Incomplete sentence

Figure 5 – What happened to rabbits after 2017?

Line 235 – To be honest, I think there is a huge chance that this seemingly correlation is just coincidental. This is a very weak and too circumstantial evidence. The evidence presented by the manuscript is strongly against, not in favor, of climatic factors driving the temporal variation of the two diseases in the analyzed years.

The authors should not highlight this observation and it definitely should not be in the abstract. The focus here must be on the result that no influence of climate factor was found on the number of cases.
In the current form, the reader might feel as if not finding this expected correlation between climate and disease occurrence is a flaw of the study. It is not. If the data and the analyzes are appropriate (and as far as I can tell, they are) this is a meaningful and important result. Finding the absence of a given pattern is a result as important for the advance of science as detecting the pattern.

Not requiring positive result is one of the most beneficial features of Peer J. I suggest the author to take full advantage of that, and advocate for the validity of the results they present.

Line 252 – The “together with mosquitoes” came out of the blue.
I recommend the authors to use the first paragraph of discussion to interpret their main findings and their implications.

Line 274 – Ambiguous sentence. Not sure whether the author mean that the LB numbers in winter is high compared to the TBE numbers in winter or to the LB numbers in summer.

Line 334 to 340 – I did not understand the explanation. Based on the explanation, when the rodent population density is low, co-feeding should be high, thus, infecting more ticks. So, why not enough ticks are infected?

Discussion and conclusion – I suggest the authors to close the discussion with conclusions that arrive or, at least, are coherent with their findings. In fact, I feel that in many points the discussion is a bit disconnected from the study results. The author should, of course, discuss other perspectives and results on the literature, and provide possible explanations for why their results did not match their expectations, but the focus of the discussion must be their findings. This problem is most evident in the final paragraph, as the authors conclude the manuscript by extensively discussing patterns that they have not found.

---

## Round 0.2 · Minor Revisions

The authors addressed the main concerns of the reviewers. However, your revised manuscript still deserves attention. Please, provide point-to-point responses according to the comments made by Reviewer #2 in the new version of your manuscript.

·

Basic reporting

.

Experimental design

.

Validity of the findings

.

Additional comments

Since the authors replied my comments and questions and introduced some minor requested, suggested changes, I suggest this paper for publicantion in PeerJ.

Reviewer 2 ·

Basic reporting

The manuscript describes a well conducted and informative analysis of the spatial and temporal variation of two diseases, Tick-Borne Encephalitis (TBE) and Lyme Borreliosis (LB), in Germany and some possible underling drivers. The manuscript is well written and clear, and figures are nice. The authors provide the necessary background for the study, so that, even non-specialists in the system can understand it.

Experimental design

The study is well designed, the research question is well defined and relevant.
As far as I can tell, the data is appropriate for the question addressed, the analysis is well conducted, resulting in meaningful results.

Validity of the findings

The issues I pointed in my previous revision were solved.
The authors present sounding analyses and results, and draw appropriate conclusion from them. The findings are valid, relevant and deserves to be published.

Additional comments

Line 114 - “rises” or “increases”, but not both

Line 139 - This sentence seems weird. Probably some mistake here.
The introduction is well written and accessible for a broad scientific audience. All the important information is provided. The last paragraph included worked well and the other problems I pointed in my last revision were solved.

Line 150 - “diseases”

Figure 3 - In the maps, it is very hard to distinguish the absence of data from the absence of cases. I suggest the authors to use another color (for instance, white) to represent the absence of data.

Line 229 - Why some federal states that has data for LB infections in some years, such as Bavaria, were indicated as “no data” in figure 4b? The authors used some criterium based on the number of years available? This must be clearer.

Line 260 - These results are of interest, but it is hard to take any lesson from absolute values. Results would be much more informative if the authors provided a comparison against the gender and age structure of the population.
Are males more infected by TBE than females just because there are more males than females on the population? Or the male individuals have an increased chance of being infected?

Line 307 - Repeated words: “in years in years”.

Discussion - The discussion is well written and accurate, however, in my view, it remains too long and a bit unfocused. The authors dedicate several large paragraphs to discuss possible drivers of the diseases that were not detected by their study.
I believe that reducing and focusing would benefit the discussion. However, as the discussion is accurate, this criticism should be seem as a suggestion to improve the paper readability, not as a barrier for publishing it.

---

## Round 0.3 · accepted · Accept

The authors have satisfactorily responded to all questions and made the necessary changes to the manuscript.